# Charge and exciton dynamics of OLEDs under high voltage nanosecond pulse: towards injection lasing

Viqar Ahmad [1,2,6], Jan Sobus[1,2,6✉], Mitchell Greenberg[3,6], Atul Shukla [1,2], Bronson Philippa [3], Almantas Pivrikas[4], George Vamvounis [3], Ronald White[3], Shih-Chun Lo [2,5✉] & Ebinazar B. Namdas [1,2✉]

Electrical pumping of organic semiconductor devices involves charge injection, transport, device on/off dynamics, exciton formation and annihilation processes. A comprehensive model analysing those entwined processes together is most helpful in determining the dominating loss pathways. In this paper, we report experimental and theoretical results of Super Yellow (Poly(p-phenylene vinylene) co-polymer) organic light emitting diodes operating at high current density under high voltage nanosecond pulses. We demonstrate complete exciton and charge carrier dynamics of devices, starting from charge injection to light emission, in a time scale spanning from the sub-ns to microsecond region, and compare results with optical pumping. The experimental data is accurately replicated by simulation, which provides a robust test platform for any organic materials. The universality of our model is successfully demonstrated by its application to three other laser active materials. The findings provide a tool to narrow the search for material and device designs for injection lasing.

[1] School of Mathematics and Physics, The University of Queensland, Brisbane, QLD 4072, Australia. [2] Centre for Organic Photonics and Electronics, The University of Queensland, Brisbane, QLD 4072, Australia. [3] College of Science and Engineering, James Cook University, Douglas, QLD 4811, Australia. [4] School of Engineering and Information Technology, Murdoch University, Murdoch, WA 6150, Australia. [5] School of Chemistry and Molecular Biosciences, The University of Queensland, Brisbane, QLD 4072, Australia. [6] These authors contributed equally: Viqar Ahmad, Jan Sobus, Mitchell Greenberg. ✉email: j.sobus@uq.edu.au; s.lo@uq.edu.au; e.namdas@uq.edu.au

Organic light-emitting diodes (OLEDs) are unarguably the most successful member of a rapidly growing family of devices based on organic semiconducting materials (including organic field-effect transistors, light-emitting field-effect transistors, organic solar cells, organic photodetectors and organic lasers). The simple OLED architecture has also proven to be a robust testing ground for a multitude of potential applications such as displays, lighting and lasers[1,2]. There is a consensus in the scientific community that lasing in such a device can only be driven in short and high intensity pulses spaced by relatively long intervals to avoid triplet accumulation and excessive Joule heating[3,4].

In optically pumped organic lasers, the exciton generation response to the optical excitation pump pulse is almost linear (assuming high enough absorption and no bleaching/multi-photon effects). That is, almost all pumped photons are absorbed and form the same number of singlet excitons, with the absorption process occurring almost instantly (on a sub-ps scale). It is, therefore, a common practice to convert pump energy to photon number as an initial stage ($t = 0$), in modelling the exciton population using rate equations[5,6].

Unfortunately, these assumptions are not valid for electrically pumped organic lasers. The transient current response to the applied voltage pulse, charge transport in the organic layer and the formation of excitons are all strongly non-linear and occur on timescales ranging from sub-ns to μs range, which are similar to the timescales of the excited state lifetimes[7–9]. Therefore, the exciton's temporal and spatial distribution is significantly altered compared to optical pumping. Hence, the device response at each step needs to be understood and solved in time domain, with solutions of the previous step being input to the next one.

In this work, we report both experimental and theoretical results of Super Yellow (SY) OLEDs operated at a high current density of 200 A cm$^{-2}$ with a pulse duration of 300 ns. We quantitatively determined the decay rate of SY OLEDs including singlet exciton decay, triplet exciton decay, singlet–singlet annihilation (SSA) rates, field-induced quenching, and electron and hole mobilities and then developed a comprehensive theoretical analysis that incorporates a simple resistance—capacitance (RC) circuit, drift-diffusion model and rate equations for exciton generation and annihilation processes, which validated experimental findings. This model revealed a narrow time frame for lasing in which the singlet density peaks under electrical pumping. In order to show that our model universally describes OLEDs under high current densities, we show its application to devices manufactured with three other common emitters—F8BT [poly (9,9-dioctylfluorene-*alt*-benzothiadiazole)], PFO (polyfluorene) and BSBCz (4,4′-bis[(*N*-carbazole)styryl]biphenyl). Importantly, the analysis uncovered universal guidelines for material and device design that enables the potential to access the injection lasing threshold. This result is significant for the realisation of injection lasing in organic materials.

## Results

**Transient current voltage analysis of OLEDs with RC circuit.** We fabricated small area (0.1–0.75 mm$^2$) OLEDs based on the emissive polymer, SY[10]. The experimental details of device fabrication and the test setup can be found in 'Methods' and Supplementary Information. Chemical structure of SY along with OLED structure and corresponding energy levels of the device are shown in Supplementary Fig. 1a–c. The OLEDs were initially tested under DC input to establish that the devices are working at optimum performance. The results of DC measurements including brightness, current density, external quantum efficiency and current efficiency can be seen in Supplementary Fig. 2a, b.

For pulse measurements, OLED devices were tested for a range of voltages from 20 to 100 V at 300-ns pulse width. Such choice of voltage range and pulse width is dictated by the scope of this work. Staying in this voltage range provides strong current and brightness signal, yet it does not have a significant effect on device degradation (Supplementary Fig. 3). Three hundred nanosecond pulse width is a time window long enough to enable simulation of exciton dynamics. Nevertheless, with the future work of reaching injection threshold in mind, we tested capabilities of our setup with shorter pulses (15 ns) of higher magnitude (up to 450 V)—Supplementary Fig. 4.

The transient response and device model for OLEDs under voltage pulse operation provides crucial insights to explain the device characteristics[7–9]. A typical transient current characteristic of SY OLEDs driven with 300 ns voltage pulse is shown in Fig. 1. A current spike occurs almost immediately after pulse onset, followed by the current relaxation to a certain steady-state value as shown in Fig. 1. At the end of the voltage pulse, negative current spike occurs before relaxing to zero amps[11]. It has also been observed that the steady-state value of the current during pulse is equal to the current observed in the DC operation mode for the same excitation voltage[12]. This current response can be explained if one models the device with the equivalent circuit presented in Fig. 1b, with $R_s$ being the series resistance (corresponding to the resistance of the contacts, electrode leads etc.) and layer resistance ($R_{layer}$) and capacitance ($C$), forming the RC element describing the organic layer[13]. The current response of such circuit is given by the following Eqs. (1) and (2):

$$V_c(t) = V_{in}(t) - R_s I(t), \tag{1}$$

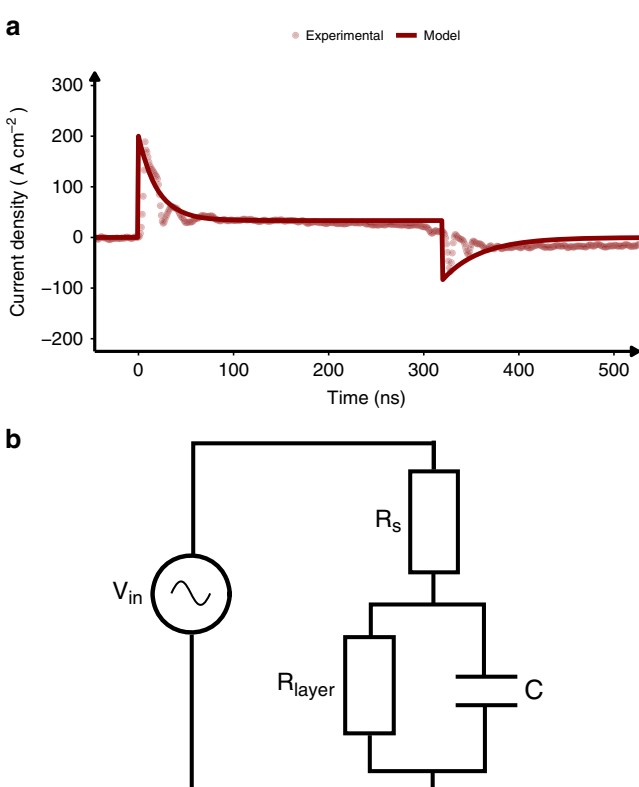

**Fig. 1 Electrical circuit model and current response of OLED.** Typical current density curve of Super Yellow OLED depicting rapid current spike and subsequent relaxation to steady state followed by negative spike at turn-off for. The OLED was excited with 300-ns voltage pulse. Fig. 1b shows equivalent circuit used for OLED modelling consisting of layer resistance ($R_{layer}$), series resistance ($R_s$), and capacitor ($C$).

$$C\frac{dV_c(t)}{dt} + V_c(t)\left(\frac{1}{R_{\text{layer}}} + \frac{1}{R_s}\right) - \frac{V_{\text{in}}(t)}{R_s} = 0, \qquad (2)$$

where $V_{\text{in}}(t)$ is the input voltage pulse, $I(t)$ is the current flowing through the device, $V_c(t)$ is the voltage applied on the OLED electrodes and $R_{\text{layer}}$ and $R_s$ are the resistances.

While these equations can be solved numerically for any input pulse $V_{\text{in}}(t)$, a wealth of information can be extracted from analytical solutions of simple pulse shapes. The solution for a square pulse is shown in Supplementary Note 1. There is an initial spike of high current (Fig. 1), which then decays according to an exponential time constant, $\tau = \frac{CR_sR_{\text{layer}}}{R_s+R_{\text{layer}}}$. This characteristic time depends on all circuit parameters and, even keeping the parameters of organic layer intact, it can be further reduced by employing small area pixel ($C$ reduction). In addition, indium tin oxide (ITO) contacts can be widened or their material can be changed ($R_s$ reduction). Obtaining $\tau$ as low as possible is especially important when the width of the electrical excitation pulses is in the ns regime. In this case, the rise and fall time of the pulse generator cannot be neglected anymore, and square pulse solution is insufficient. When $\tau$ becomes comparable to the pulse generator rise/fall time, the more suitable analytical solution is that of the trapezoid pulse—we show solution for this case in Supplementary Note 2. The rise and fall time of the input voltage significantly affects the shape of the current transient, as can be seen in Supplementary Fig. 5a, b.

It is worth noting that using a circuit approach to analyse the full device enables other methods—like impedance spectroscopy—to be used as a tool to confirm the results obtained from the pulse data[10]. Both the values of the resistance (constant $R_s$ and $R_{\text{layer}}$ as a function of applied voltage) as well as the capacitance can be extracted (see Supplementary Fig. 6). Extracted $R_s$, $R_{\text{layer}}$ and capacitor values allow the calculation of time constant ($\tau$), which was found to be $(8 \pm 0.5)$ ns for 0.3-mm$^2$ OLEDs, which is comparable to that of $\approx$10 ns obtained by fitting the experimental current pulse as shown in Supplementary Table 1. The time constant obtained by fitting the current response is slightly longer due to noisy current response affecting the fitting.

**Electrical simulation using drift-diffusion model.** Analysis of OLED as a simple RC circuit can bring helpful insights into the optimal device architecture and electrical response dynamics. However, this approach is too simplistic to describe the non-linear processes occurring in the organic layer itself. To properly predict and model exciton dynamics, a drift-diffusion model[14] is required, described in detail in Supplementary Note 3. The drift-diffusion model includes drift and diffusion of electrons ($n$) and holes ($p$), the influence of space charge on the electric field, temperature-dependent generation of free charge carriers and Langevin radiative recombination rate ($\beta$). This model was used to fit the current transients obtained for SY devices at different pulse voltages and pulse lengths, where the experimental and simulation-fitted parameters given in Table 1 were used across the range of pulse voltages. We have also simulated the time when the peak exciton formation (charge carrier recombination) takes place as a function of injected current and emitter mobility to illustrate its temporal position and width (Supplementary Fig. 7). It can be seen that, depending on the material performance and driving conditions, it can be as fast as deep in the sub-nanosecond regime (with a well-defined peak position) or as slow as hundreds of nanoseconds (with the tail extending to µs regime).

The simulation accurately predicts the experimental transient current as shown in Fig. 2a. We want to highlight that the fitted values in Table 1 are close to the experimental and literature values of rate constants that were used as the starting values for the fit, thus confirming accuracy of our model. The simulated current shows that that the model captures the decay in current for a wide range of voltages (60–100 V). However, the model does not capture some additional behaviour at very short timescales. It is possible that a mismatch of characteristic impedances in the measurement circuit is introducing additional effects, which we have not attempted to simulate. Second, the device recovery after the voltage pulse is turned off is simulated to be much faster than experimentally observed, which we attribute to charges being released from traps in a way that is not fully accounted for in the model. Nevertheless, the device current density for the majority of the pulse is accurately captured—summary of simulated and measured current densities (initial peak and steady state) is shown in Supplementary Fig. 8. Peak current density is linearly dependent on voltage in agreement with circuit model.

**Exciton dynamics and annihilation processes.** After the temporal and spatial charge carrier density solutions are obtained, one can use the bimolecular recombination as the generation term for excitons. Usually, either a fixed number of excitons (pulse energy divided by single-photon energy in case of pumping with short laser pulses)[5,6] or current density over the thickness of recombination layer (for electrical pumping)[6,11,15] are used in this place. We emphasise that such an approach is not valid, since, first, holes and electrons will have different spatial and temporal concentration profiles in the organic layer. Second, even in steady-state conditions, the recombination has a chance to occur only in the volume where these concentrations overlap. We extracted the maximum $\beta np$ generation term from our model and compared it to the commonly used $J/ed$ approximation—Supplementary Fig. 9 (where $J$ is current density, $e$ is electron charge and $d$ is the emitter layer thickness). One can see that this approximation is only valid at low current densities and strongly overestimates the exciton generation term for $J > 100$ A cm$^{-2}$. Our approach has several advantages over methods just mentioned. It considers non-uniform charge and field distribution across the width of the organic layer as well as time needed for charge carriers to recombine, forming excitons. The equations governing the spatial and temporal populations of singlets (S) and triplets (T) are given by Eqs. (3) and (4), respectively.

$$\frac{dS}{dt} = 0.25\beta np - k_S S - k_{\text{ISC}}S + k_{\text{RISC}}T - k_{\text{SSA}}S^2 - k_{\text{STA}}ST \\ + 0.25k_{\text{TTA}}T^2 - k_{\text{SPA}}(n+p)S - k_{\text{FQ}}(E)S, \qquad (3)$$

$$\frac{dT}{dt} = 0.75\beta np - k_T T + k_{\text{ISC}}S - k_{\text{RISC}}T - k_{\text{TTA}}T^2 \\ - 1.25k_{\text{TTA}}T^2 - k_{\text{TPA}}(n+p)T - k_{\text{FQ}}(E)T. \qquad (4)$$

Here, the pre-factors 0.25 and 0.75 in front of the generation terms come from the spin statistics governing exciton formation. $k_s$ is the fluorescence decay rate, $k_T$ is phosphorescent decay rate, $k_{\text{ISC}}$ and $k_{\text{RISC}}$ are intersystem crossing (ISC) and reverse intersystem crossing (RISC) rates, respectively ($k_{\text{RISC}} = 0$ in this system but significant in thermally activated delayed fluorescence molecules). $k_{\text{SSA}}$, $k_{\text{STA}}$, $k_{\text{TTA}}$ are bimolecular recombination rates for SSA, singlet–triplet (STA) and triplet–triplet annihilation (TTA)[6]. Table 1 summarises the experimentally measured $k_S$, $k_{\text{ISC}}$ and $k_T$ and $k_{\text{SSA}}$ of SY films. The bimolecular recombination rates for SSA were measured using transient pump fluences method (see Supplementary Fig. 24). The $k_{\text{STA}}$ and $k_{\text{TTA}}$ were taken from literature and listed in Table 1. We note that all these transient processes are present in the system for optical and electrical pumping alike. However, in the presence of charge carriers and

**Table 1 Decay and annihilation rates.**

| Decay rates ($s^{-1}$) | Experimental value | Fitted value |
|---|---|---|
| Singlet decay rate ($k_S$) | $3 \times 10^8$ | $3 \times 10^8$ |
| Intersystem crossing ($k_{ISC}$) | $2 \times 10^8$ | $2 \times 10^8$ |
| Triplet decay rate ($k_T$) | $6 \times 10^5$ | $6 \times 10^5$ |
| Annihilation rates ($cm^3 s^{-1}$) | Initial | Fitted |
| Singlet–singlet ($k_{SS}$) | $2.9 \times 10^{-10}$ (measured) | $7 \times 10^{-10}$ |
| Singlet–triplet ($k_{ST}$) | $1 \times 10^{-10}$ (ref. [21]) | $1.8 \times 10^{-10}$ |
| Triplet–triplet ($k_{TT}$) | $1 \times 10^{-13}$ (ref. [22]) | $6.5 \times 10^{-13}$ |
| Singlet–polaron ($k_{SP}$) | $5 \times 10^{-14}$ (ref. [23a]) | $5 \times 10^{-14}$ |
| Triplet–polaron ($k_{TP}$) | – | $4 \times 10^{-13}$ |
| Charge carrier dynamics | | |
| Exciton-field separation constant ($k_{FQ}$) | $465\ s^{-1}$ (measured) | $850\ s^{-1}$ (fitted) |
| Exciton-field separation Bessel term ($k_{FQ}$) | $1 \times 10^7\ V\ m^{-1}$ (measured) | $2 \times 10^8\ V\ m^{-1}$ (fitted) |
| Device permittivity | $3.4 \pm 0.2$ (measured) | |
| Active layer thickness | $60 \pm 3$ nm (measured) | |
| Hole mobility | $5.0 \times 10^{-5}\ cm^2\ V^{-1}\ s^{-1}$ (measured) | $7.0 \times 10^{-5}\ cm^2\ V^{-1}\ s^{-1}$ (fitted) |
| Electron mobility | $1.0 \times 10^{-5}\ cm^2\ V^{-1}\ s^{-1}$ (measured) | $1.4 \times 10^{-5}\ cm^2\ V^{-1}\ s^{-1}$ (fitted) |
| Electron steady-state mobility | $\frac{\mu_{ss,n}}{\mu_{0,n}} = 0.77$ (fitted) | |
| Hole steady-state mobility | $\frac{\mu_{ss,p}}{\mu_{0,p}} = 0.13$ (fitted) | |
| Electron mobility decay rate | 256 ns (fitted) | |
| Hole mobility decay rate | 92 ns (fitted) | |

Decay and annihilation rates of various processes with carrier dynamics of the device.
[a]An estimate of singlet-polaron annihilation rate ($k_{SP}$) can be found by the following equation[23]: $k_{SP} = \frac{f_Q}{n_h f_R \tau_R}$,
where $f_Q/f_R$ is the fraction of quenched to radiated holes, $n_h$ is the volume concentration of holes and $\tau_R$ is the singlet radiative decay rate. For a hole density of $10^{19}\ cm^{-3}$, radiative decay rate of 2 ns and fraction of quenched to radiated holes being $1 \times 10^{-3}$, the singlet-polaron annihilation rate becomes $5 \times 10^{-14}\ cm^3 s^{-1}$.

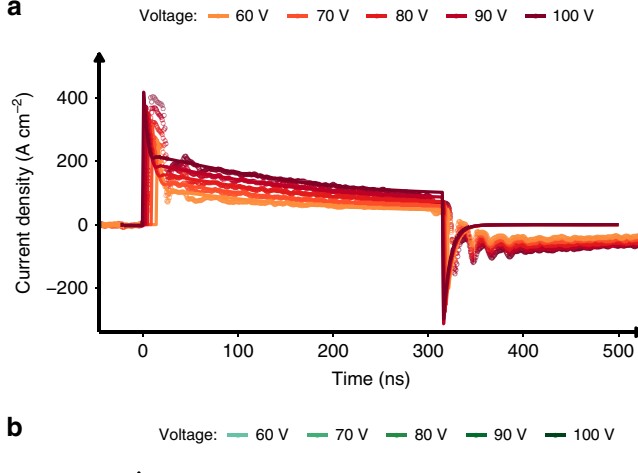

**a** Current.

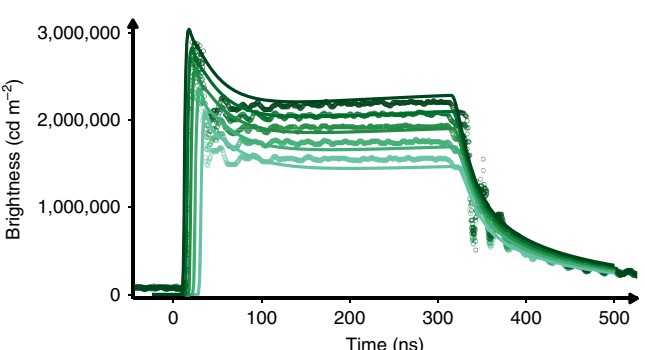

**b** Brightness.

**Fig. 2 Simulated and experimental OLED response to pulse input.**
**a** Current. **b** Brightness. Simulated and experimental response of the OLED subjected to the 300-ns voltage pulses with amplitudes varying from 60 to 100 V.

electric fields, additional decay pathways for excitons are present, and it is necessary to account for singlet-polaron annihilation (SPA) given by rate ($k_{SPA}$), triplet-polaron annihilation (TPA) given by rate ($k_{TPA}$) and field quenching[16] ($k_{FQ}$) processes. The rate constants for field quenching[16], singlet-polaron and triplet-polaron quenching were measured using field-dependent PL quenching experiments for hole and electron-only devices (see Supplementary Note 4). SSA rate was also experimentally developed with details in Supplementary Note 5.

Simulated exciton dynamics using above experimental parameters are presented in Fig. 2b. The simulated brightness is in excellent agreement with the experimental data. The nearly flat shape of the brightness response (despite a large reduction in current) is captured in the model, as is the relative brightness at each of the simulated voltages. It should be emphasised that the same set of parameters, given in Table 1, were used for all simulations and it is only the applied voltage that was adjusted.

## Discussion

To gain further insights into injection lasing, we have simulated the evolution of exciton and polaron densities for SY emitter over time under varying pumping conditions. Figure 3a–d shows calculated singlet, triplet, polaron and charge carrier densities in SY emissive layer (averaged over volume) from 1 ps to 1 µs and current densities (*J*) from 100 to 100 kA cm$^{-2}$. Independent of *J*, all plots share similar features—initial growth of electron and hole populations, well-defined singlet population peak and polaron/triplet populations dominating at longer times with concentrations two orders of magnitude higher than the singlets. However, there is a noticeable change in the temporal position of the singlet peak, which strongly depends on the injected current density, being pushed to shorter times as *J* increases. There is also a noticeable difference in the peak-to-steady-state singlet ratio, which increases with growing *J*. It is clear that in order to access the highest singlet density, a narrow window between too short (charge carriers did not manage to recombine yet) and too long (quenching processes killing singlet population) times exists. This

 

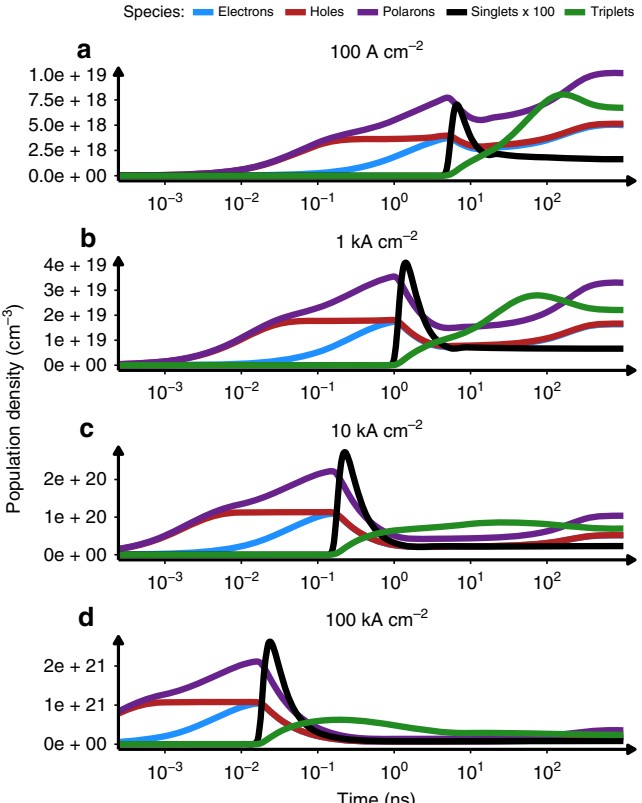

**Fig. 3 Population transients under electrical excitation.** Simulated temporal concentrations (averaged over recombination zone) of charge electrons, holes, polarons, singlets and triplets in devices under electrical pumping in the range of current densities, **a** 100 A cm$^{-2}$. **b** 1000 A cm$^{-2}$. **c** 10,000 A cm$^{-2}$. **d** 100,000 A cm$^{-2}$.

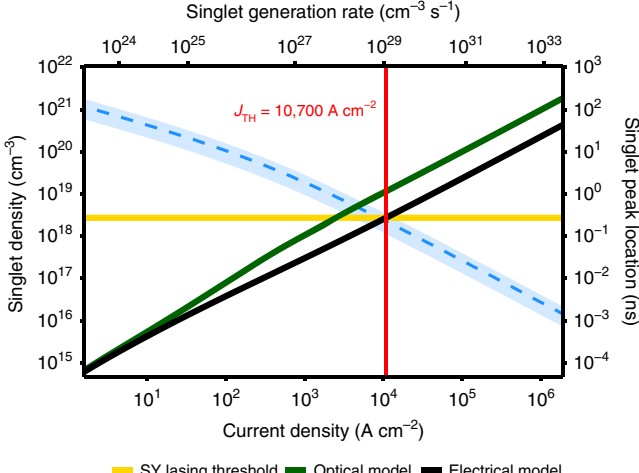

**Fig. 4 Maximum singlet density as a function of pumping current density.** Singlet concentrations obtained from the optical (green) and electrical (black) models are shown. Dashed blue line indicates time after pulse onset, at which the maximum was achieved. SY lasing threshold (yellow) expressed in singlet density and corresponding current density threshold (red) included for reference.

is a significant difference from the optical pumping case, where shorter pulses are more effective, in general. In addition, the width recombination zone evolves as well (Supplementary Fig. 10). At the time the singlet population reaches maximum, recombination zone spans around 20% of the total organic layer width and the singlet spatial concentration is highly non-uniform, with peak values being order of magnitude higher than the averaged one (Supplementary Figs. 10 and 11). At longer times, it extends over the whole volume of organic layer, and the singlet population spreads out uniformly.

To put obtained singlet concentrations into perspective, we compared them to the reported lasing threshold for SY, which is nearly 75.6 kW cm$^{-2}$ for a 5-ns pulse and an excitation wavelength of 450 nm[17]. In order to find the current density needed to achieve injection lasing, 75.6 kW cm$^{-2}$ (i.e., 378 μJ cm$^{-2}$, $8.55 \times 10^{14}$ photons cm$^{-2}$) must be converted to the number of absorbed photons per cm$^3$, per second. As a rough estimate, assuming uniform absorption for 5 ns in an active layer with 250-nm thickness[17], the number of absorbed photons (and thus maximum singlet formation rate) translates to $6.85 \times 10^{27}$ photons per cm$^3$ per second. We built an 'optical' version of our model, where the triplet generation term is equal to zero (only source of triplets is ISC) and polaron terms do not exist. Under such conditions, this singlet formation rate results in singlet concentration of $2.7 \times 10^{18}$ cm$^{-2}$, which we use as a lasing threshold singlet concentration for SY. We then use the full, electrical model to evaluate what current density is required to reach the same singlet formation rate and what will be the resulting singlet concentration with triplet generation and quenching terms present. Even in the most optimistic case,

assuming we can access the temporal maximum of singlets (Fig. 4), the obtained electrical model threshold (10.7 kA cm$^{-2}$, indicated by red line on the plot) is around five times higher than the value derived directly from the optical model (2.3 kA cm$^{-2}$). This underlines the importance of using dedicated electrical model in order to avoid underestimation of injection lasing threshold. Moreover, the time at which the maximum singlet density is obtained (dashed blue line, 220 ps for 10.7 kA cm$^{-2}$) lies beyond the capabilities of available high-voltage pulse generators, which typically produce pulses in the range of 5–100 ns. Singlet densities probed at that time range (Supplementary Fig. 12) show much gentler slope with increased $J$, resulting in threshold current densities of over $10^6$ A cm$^{-2}$. The maximum steady-state current density (required for exciton generation) that has been achieved in this report is ~200 A cm$^{-2}$ that is well below SY's thresholds, although, the peak brightness achieved in this work is above 2 million cd m$^{-2}$.

One way of achieving a higher current density could be to have device areas even smaller than 0.1 mm$^2$, used in this work. Other way would be to switch to active material with lower lasing threshold and/or reduced quenching processes, which would extend the time of peak singlet concentration and make it more accessible. In order to identify good candidates for such material, we checked the contributions of different decay pathways in the total singlet depopulation (Fig. 5a–d). One can see that bimolecular processes (SSA at singlet temporal peak and STA at longer times) are main competitors to the fluorescence, with their contributions even more profound at higher $J$. Even at modest 1-kA cm$^{-2}$ SSA and STA combined consume over 90% of singlets at 1 ns already (Supplementary Fig. 13). While SSA is hard to avoid, STA can be minimised by choosing materials with appropriate singlet and triplet energy spectra, thus minimising $k_{STA}$. Finally, power dissipation issues must be considered. We compared the optical power density (W cm$^{-2}$) and electrical power (product of voltage and current density) necessary to obtain the same exciton generation rate—Supplementary Fig. 14. It is apparent that even if the dream material with low threshold population (and corresponding low threshold generation rate) is found, delivered power flux will be much harder to dissipate than in the optical case.

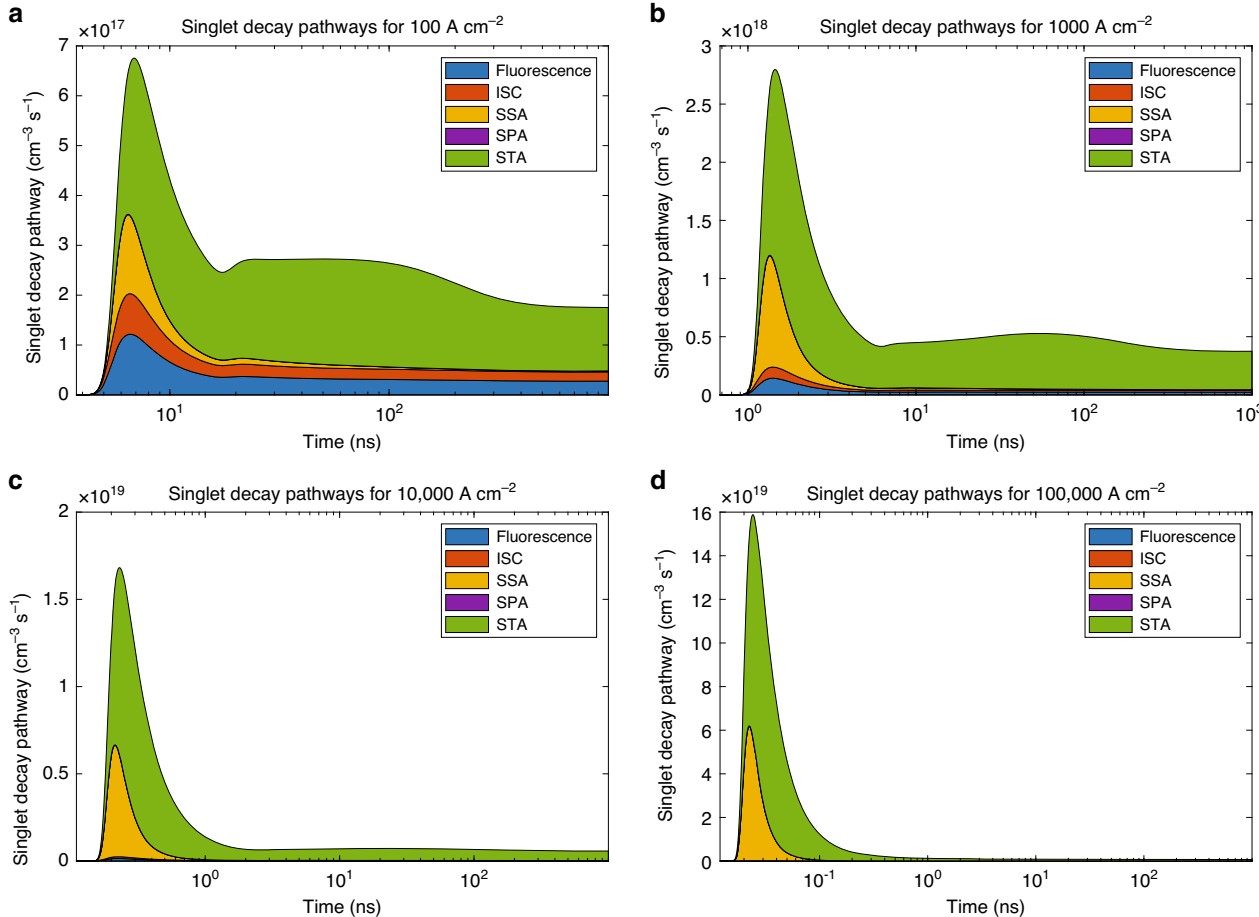

**Fig. 5 Contributions of various singlet decay pathways plotted versus time.** Here, ISC, SSA, SPA and STA stand for intersystem crossing, singlet–singlet annihilation, singlet-polaron annihilation and singlet–triplet annihilation, respectively. Results obtained for four different current densities are shown with **a** 100 A cm$^{-2}$. **b** 1000 A cm$^{-2}$. **c** 10,000 A cm$^{-2}$. **d** 100,000 A cm$^{-2}$.

While we performed the full analysis for SY, it is important to remember that our model is applicable to any organic material subjected to high current densities. In order to prove that, we manufactured OLEDs with active layers composed of F8BT and PFO. Supplementary Figure 15a, b shows the transient current and EL response of F8BT, and Supplementary Fig. 15c, d shows transient current and EL response of PFO OLEDs, respectively, driven with 300 ns at variable voltage pulses. The devices once again showed excellent current–voltage-brightness characteristics with peak brightness exceeding 1 million cd m$^{-2}$. To complete the picture, we included the current–voltage and current efficiency data for F8BT, PFO and SY (as reference) in the form of Supplementary Fig. 16. It combines both, the DC and pulse regime to show full evolution of device characteristics. In addition, since the efficiency roll-off is evident in Supplementary Fig. 16, we simulated the contributions of different exciton decay pathways as a function of injected current density to identify main loss mechanisms in these materials—Supplementary Fig. 17. One can see that while for SY the dominating loss mechanism is STA, for F8BT, it is SSA. In PFO both mechanisms show impact within the same order of magnitude.

Simulated transient current–voltage-brightness dynamics of both F8BT and PFO using our model are also presented in Supplementary Fig. 15a–d. The simulated brightness and current densities are in excellent agreement with the experimental data. This agreement is an important result as it shows the universality of our model and approach across a number of light-emitting polymers. Supplementary Tables 2 and 3 summarise the rate constants of F8BT and PFO neat films, primarily taken from the literature, with fitted values alongside. Electron and hole mobilities of neat film of PFO were measured experimentally from hole-only and electron-only devices.

From these simulations, we can further estimate the electrical lasing threshold required for F8BT and PFO lasers. The reported optical lasing thresholds for F8BT and PFO are 4.1 µJ cm$^{-2}$ at 488 nm (100 fs) pulse[18] and 29 µJ cm$^{-2}$ at 390 nm (0.5 ns) pulse excitation[19], respectively. This translates to a singlet density at lasing threshold of ≈10$^{19}$ cm$^{-3}$ and ≈5 × 10$^{17}$ cm$^{-3}$ for F8BT and PFO, respectively. Supplementary Fig. 18a, b shows simulated singlet as a function of current density for optical (only triplets as quenching) and electrical (all quenching terms) models for F8BT and PFO, respectively. The electrical model predicts thresholds of F8BT and PFO to be around 7.7 × 10$^4$ A cm$^{-2}$ and 2.2 × 10$^3$ A cm$^{-2}$, respectively (indicated by red line), which is well above the experimentally achievable current densities (≈200 A cm$^{-2}$). These results highlight the importance of using dedicated electrical model to avoid underestimation of injection lasing threshold.

In addition, to show applicability of our model to small-molecule-based OLEDs, we simulated BSBCz-based devices reported in a recent work by Adachi[20] group using rate constants reported for that emitter in the literature (Supplementary Table 4). The simulated current threshold value of 750 A cm$^{-2}$ matches the reported experimental value of around 600 A cm$^{-2}$ pretty well, thus confirming broad application scope of our model (Supplementary Fig. 19). Furthermore, we show that the light outcoupling channels are more dependent on the devices

structure than the emitter used—which is why we employ the same device structure in all experiments (Supplementary Fig. 20 and Supplementary Note 6). It is important to note that past lasing studies were focus on fluorescent emitters, and often the research considers the properties of the emitter itself as opposed to an entire OLED device stack. Furthermore, high-performing OLED devices are mainly based non-laser active (phosphorescent and TADF) emitters and evaporated fluorescent small molecules with moderate lasing threshold. In contrast, when it comes to lasing, materials with champion optical lasing thresholds are coming from a family of oligomers and polymers (Supplementary Fig. 21 and Supplementary Tables 5, 6) with modest OLED performance. Therefore, universality of our model across all materials provides a powerful tool to narrow down the search for both material and device designs for injection lasing.

## Conclusions

To conclude, we have successfully developed an all-inclusive electrical and optical model of OLEDs based on SY as the emitter, and demonstrated complete transient exciton and polaron dynamics starting from charge injection to light emission. The OLED exhibited transient brightness of $\approx 2$ million cd m$^{-2}$ at a current density of 200 A cm$^{-2}$. The model fits well with experimentally obtained data, demonstrating viability of the presented work. Furthermore, we successfully demonstrated the applicability of our model to three other common emitters, F8BT and PFO as well as reported data for small-molecule-based BSBCz diodes. In addition, we demonstrated that, contrary to optical pumping, there is a narrow time window to access peak singlet density and we provided guidelines both for device design and material choice that can lead to injection lasing. We illustrated how direct adaptation of methods used in optical pumping leads to overestimation of exciton generation and underestimation of threshold current density. Our results provide holistic approach to understanding the response of organic materials to pulsed electrical excitation. We believe that these results are of significant importance for the field of organic semiconductor laser research and can be treated as a platform to model other laser-active emitters in future.

## Methods

**Device fabrication details**. OLED design involved the use of glass substrate with patterned ITO. The cleaning process included heating the substrate in deionized water with Alconox at 150 °C for 15 min. Alconox treatment was followed by sonication in deionized water for 5 min. The substrates were further sonicated in acetone and isopropanol for 5 min each. The cleaning process culminated with a 30-min ultraviolet–ozone surface treatment.

A 30-nm layer of PEDOT:PSS was used on top of the cleaned substrate to serve as a hole injection layer. PEDOT:PSS was spin-coated at 5000 rpm for 60 s and annealed at 125 °C for 20 min in air. SY (PDY-132, Sigma Aldrich) was used as the emissive layer (prepared with 7 mg mL$^{-1}$ in toluene). A 60-nm-thick SY layer was achieved using a 3000-rpm spin for 60 s, followed by a 50 °C anneal for 20 min. Barium and aluminium were used as the top contacts and were deposited via evaporation using a shadow mask having a thickness of 7 and 100 nm, respectively. PFO and F8BT (1 Material Inc.) were dissolved in toluene (7 mg mL$^{-1}$). Similar architecture as SY OLEDs was employed for F8BT and PFO OLEDs as well. Details of electron and hole-only devices along with details of quenching experiment and determination of rate constants can be found in Supplementary Figs. 22–24 and Supplementary Table 7.

**Pulse-measurement setup**. Pulse measurements were performed using AVTECH pulse generators, AV10111B1-B and AVRK-3-B, which have a rated rise and fall time of <6 ns each. The pulse widths applied ranged from 6 to 300 ns. Low loss BNC cables were used to minimise the effects of ringing due to fast switching (PE-SR402FLJ, Pasternack®). Agilent Technologies digital storage oscilloscope was used with 2.5-GHz frequency range and 2.25-Gs s$^{-1}$ sampling rate to observe current and electroluminescence (EL) response. EL response was collected using a calibrated Hamamatsu Photonics photomultiplier H10721-01 having a rise time of 0.57 ns. Current through OLED was extracted using fast current probe from Integrated Sensor Technology (711 UHF) with a 7.5-mV mA$^{-1}$ sensitivity. The pulse-measurement setup is shown in Supplementary Fig. 25.

## Data availability

The data that support the findings of this study are available from the authors upon reasonable request.

## Code availability

The code used for simulations used in this study is available from the authors upon reasonable request.

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

## Acknowledgements

We thank the Australian Research Council (DP160100700) and Department of Industry, Innovation and Science (AISRF53765) for financial support. V.A. was supported by Australian Government's Australian Postgraduate Award (APA) scholarship. MG was supported by an Australian Government Research Training Program scholarship and A.S. was funded by the UQ's Research and Training Program. This work was performed in part at the Queensland node of the Australian National Fabrication Facility Queensland Node (ANFF-Q)—a company established under the National Collaborative Research Infrastructure Strategy to provide nano- and micro-fabrication facilities for Australia's researchers.

## Author contributions

J.S. and E.B.N. conceived the idea of the manuscript. V.A. and J.S. designed and built high-voltage pulse setup. V.A. manufactured devices and collected all experimental OLED data. A.S. performed rate constant measurements. J.S. developed circuit model and impedance analysis. B.P. developed and implemented drift-diffusion model. M.G., B.P., A.P. and J.S. expanded theoretical model to include exciton dynamics. M.G. and B.P. fitted the experimental data with the full model and generated the simulated results. V.A., J.S., M.G., B.P. and E.B.N. drafted the manuscript. G.V., R.W., S.-C.L. and E.B.N. supervised the work. All the authors contributed to data analysis and discussion of the results.

## Competing interests

The authors declare no competing interests.
