## [Peer Review File · Nature Communications]

Reviewers' Comments:

Reviewer #1:

Remarks to the Author:

1. This is a very interesting paper on charge and exciton dynamics in 3 common polymer LEDs (Super Yellow (SY) PPV, F8BT, and PFO) at high pulsed current, whose key findings are claimed to "provide essential protocols for material and device design for injection lasing." As expected, the authors did not observe electrically pumped lasing. And while their figures appear to show how close, or distant, their devices operate relative to the conditions required for electrically-pumped lasing, that gap is still not clear, and should be made much clearer in the text and in the figures.
2. If this work focused on highly efficient small molecule OLEDs, I would be more favorably disposed to accept it for publication in Nat. Comm. In its present form, it is difficult to see how relevant it is for highly efficient small molecule OLEDs. For example, in the PLEDs many excitons might be quenched because of their proximity to the metal cathode and losses to surface plasmons at the organic/metal cathode interface. However, it may still prove acceptable if the results could be compared to the earlier studies of the Adachi group on small molecule OLEDs at high current pulses.

Reviewer #2:

Remarks to the Author:

Investigation of injecting organic lasers is important, but great challenging. In this article, the authors carried out research on the analysis of charge and excitons dynamic in OLEDs through experiment and theoretical simulation. Based on the SY OLEDs, their results demonstrated that the complete excitons and charge dynamics starting from the charge injection to light emission, in a wide time range. They proposed theoretical model to simulate the experimental results and found good consistence, which further applied to F8BT and PFO polymers. This research is interesting, however, the following concerns should be fully addressed.

For instance, in this article, the authors mentioned that their proposed theoretical model could accurately simulate the experimental results, not only for SY OLEDs but also for F8BT and PFO polymers. The question is that as for the charge injection, transport, exciton formation and annihilation in the device operation also mentioned by authors is very complex and may be affected by many factors, such as device structures, interface contact quality, layer thickness, and also the molecular solid-state structures, etc., how the theoretical model well match these three different polymers, without through elaborate control and modulation, especially the molecular structures in solid state. And how this can be concluded to be useful for other organic materials. The assumed conditions and approximate condition in the theoretical calculation should be detailed discussed in the text.

Following above the question, the exciton decay pathways and rates are very sensitive to the experimental environment, thus it is not accurate to cite the k_{STA} , k_{TTA} and k_{TPA} values (Table 1) from literature directly, which may significantly affect the simulation results.

As tabulated in Table S2, the measured singlet decay rate value of F8BT is $1.1 \times 10^{-9} \text{ s}^{-1}$, while the fitted value is totally different as $4.5 \times 10^8 \text{ s}^{-1}$. Same for the values of PFO in Table S3. What's the reason?

The statement in the last second sentence in "Discussion" section: "The electrical model predicts threshold of F8BT and PFO to be around 10^5 A cm^{-2} and $3 \times 10^3 \text{ A cm}^{-2}$ respectively, which is well below the experimentally achievable current densities ($\sim 200 \text{ A/cm}^2$)." It is confused this sentence, below or above?

As a typical OLED for further pulse investigation, except for the current density and brightness, other direct characterization data are also crucial, such as the stability of the devices under high current, if there is fast decay for the brightness?

During the pulse measurement, how about the device performances and working conditions? The corresponding I-V curves should provided.

The authors mentioned "Electrically pumped organic lasers... the formation of excitons are all

strongly non-linear and occur on timescales ranging from sub-ns to μs range". References are needed here to prove this point.

Reviewer #3:

Remarks to the Author:

In this manuscript, Viqar A. et al. tried to study the charge and exciton dynamics of organic light-emitting diodes under high voltage nanosecond pulse: towards injection lasing, a very interesting topic in the field. The authors performed short pulse voltage to understand mechanism of exciton and charge carrier dynamics of OLEDs at high current density. In this work were well done, and the analysis of the results gives physical understanding of devices. Since this study is step for organic semiconductor material and device design for injection lasing, this reviewer would like to recommend for publication in Nature Communications, after considering the following comments.

Page-2, Abstract do not suffice to understand their role of charge and exciton dynamics of OLED at high current density. "in a time scale spanning from the sub-ns to microsecond region" There is no any explanation or experimental data for microsecond region. "200 A cm^{-2} and 2 million cd m^{-2} ", Base on author's conclusion its necessary to applied several K A cm^{-2} to get lasing. Why didn't Author try very short voltage pulse ~ 20 ; In that case they can apply high current to device.

Page 3, "at high current density of 200 A cm^{-2} with a pulse duration of 300 ns" Prof. C. Adachi group reported good approaching for electrical pumping organic lasing base on BSBCz with very high current density ca. 3 K A cm^{-2} using short voltage pulse (400 ns) (Appl. Phys. Lett.106 093301, 2015 and J. Appl. Phys. 118, 155501, 2015). It is better to compare their work, what is superior of this work comparing with Prof. C. Adachi group reported works? Could Author have try such high current to super yellow, F8BT and PFO based OLEDs?

Page 4, "The experimental details of device fabrication and the test setup can be found in Supplementary Information" this reviewer couldn't find any device fabrication information in SI Page 12, "The electrical model predicts threshold of F8BT and PFO to be around 105 A cm^{-2} and 3×10^3 A cm^{-2} respectively, which is well below the experimentally achievable current densities (~ 200 A/ cm^2)" I am not sure this is correct, Author should check carefully values

Page 12, It is well known, for Laser diodes based on organic semiconductor materials have high threshold current densities that require the suppression of various inherent loss processes. One way to study such loss processes is to analyze the EQE roll-off in OLEDs. Author should discuss EQE roll-off behavior of their device.

Reviewer #1 (Remarks to the Author):

1. This is a very interesting paper on charge and exciton dynamics in 3 common polymer LEDs (Super Yellow (SY) PPV, F8BT, and PFO) at high pulsed current, whose key findings are claimed to “provide essential protocols for material and device design for injection lasing.” As expected, the authors did not observe electrically pumped lasing. And while their figures appear to show how close, or distant, their devices operate relative to the conditions required for electrically-pumped lasing, that gap is still not clear, and should be made much clearer in the text and in the figures.

Reply: We thank the Reviewer for the valuable comment. First of all, the fact that we did not observe the injection is not surprising – our devices do not have a resonator structure and SY is not a perfect material to obtain injection lasing mainly due to its high triplet related losses. Instead, the main focus of the manuscript is to provide a model describing full electrical and optical device response. We agree, however, that the required current threshold should be indicated in a clearer way and thus we added the threshold line and value to **Fig. 4** and **Fig. S18**--see below. The following statements have now been added to provide clearer current threshold values in the text:

- “11 kA cm⁻²” has been changed to more precise “10.7 kA cm⁻²” and [current threshold] “indicated by red line” has been added on Page 11.
- Similarly, more precise values of 77 kA cm⁻² for F8BT and 2,200 A cm⁻² for PFO as well as “indicated by red line” has been added on Page 13.

2. If this work focused on highly efficient small molecule OLEDs, I would be more favourably disposed to accept it for publication in Nat. Comm. In its present form, it is difficult to see how relevant it is for highly efficient small molecule OLEDs. For example, in the PLEDs many excitons might be quenched because of their proximity to the metal cathode and losses to surface plasmons at the organic/metal cathode interface. However, it may still prove acceptable if the results could be compared to the earlier studies of the Adachi group on small molecule OLEDs at high current pulses.

Reply: We thank the Reviewer’s comment. We agree that there might be additional losses resulted from device architecture, but those are generally not limited to PLEDs (similar quenching would appear if we swap the emissive layer to a small molecule). In other words, our model is not concerned by the molecule type (unless polymeric structure introduces some additional traps states or morphological anisotropy). To illustrate its same applicability to small molecule devices we simulated the BSBCz (4,4’-bis[(N-carbazole)styryl]biphenyl) based devices reported by Adachi group¹, using the rate constant values reported in the literature (**Table S4**). The obtained values of singlet densities vs pumping strength (in optical and electrical case) are included as a new **Fig. S19** below.

Fig. S19 Simulated singlet densities in BSBCz devices reported by C. Adachi¹ subject to optical and electrical pumping and obtained threshold current density.

One can see that the current threshold value obtained from our model (750 A cm^{-2}) well matches the experimental value reported in the original paper ($\approx 600 \text{ A cm}^{-2}$), thus validating our model. We emphasise that there was no fine tuning or curve fitting of the rate constants. All model parameters were taken directly from literature. We have added the following statement in the main text:

- *“In addition, to show applicability of our model to small molecule based OLEDs, we simulated BSBCz based devices reported in a recent work by C. Adachi¹ using rate constants reported for that emitter in the literature (Table S4). The simulated current threshold value of 750 A cm^{-2} matches the reported experimental value of around 600 A cm^{-2} pretty well, thus confirming broad application scope of our model (Fig. S19).” [page 13]*

Reviewer #2 (Remarks to the Author):

Investigation of injecting organic lasers is important, but great challenging. In this article, the authors carried out research on the analysis of charge and excitons dynamic in OLETs through experiment and theoretical simulation. Based on the SY OLEDs, their results demonstrated that the complete excitons and charge dynamics starting from the charge injection to light emission, in a wide time range. They proposed theoretical model to simulate the experimental results and found good consistence, which

further applied to F8BT and PFO polymers. This research is interesting, however, the following concerns should be fully addressed.

For instance, in this article, the authors mentioned that their proposed theoretical model could accurately simulate the experimental results, not only for SY OLEDs but also for F8BT and PFO polymers. The question is that as for the charge injection, transport, exciton formation and annihilation in the device operation also mentioned by authors is very complex and may be affected by many factors, such as device structures, interface contact quality, layer thickness, and also the molecular solid-state structures, etc., how the theoretical model well match these three different polymers, without through elaborate control and modulation, especially the molecular structures in solid state. And how this can be concluded to be useful for other organic materials. The assumed conditions and approximate condition in the theoretical calculation should be detailed discussed in the text.

Reply: We thank the Reviewer for the insightful comment. We have significantly expanded our description of the model in Supplementary Section A3. It has now covered the discussion of charge injection, trapping and sample homogeneity.

Following above the question, the exciton decay pathways and rates are very sensitive to the experimental environment, thus it is not accurate to cite the k_{STA} , k_{TTA} and k_{TPA} values (Table 1) from literature directly, which may significantly affect the simulation results.

As tabulated in Table S2, the measured singlet decay rate value of F8BT is $1.1 \times 10^{-9} \text{ s}^{-1}$, while the fitted value is totally different as $4.5 \times 10^8 \text{ s}^{-1}$. Same for the values of PFO in Table S3. What's the reason?

Reply: We thank the Reviewer's valuable comment. The values in Tables S2 and S3 contained typing errors; they have been fixed to $1.1 \times 10^9 \text{ s}^{-1}$ and $4.5 \times 10^8 \text{ s}^{-1}$, respectively. We agree that the values of rate constants, while ideally only material dependent, can be affected by the environment. However, these side effects certainly shouldn't result in the variation of rate constants by orders of magnitude. We highlight that in our work we measured all the rate constants, for which we had the experimental capacity to do so. The remaining ones, taken from literature, served mostly as initial values for the rates fitted with our model. The fact that the values of rate constants obtained by fitting experimental transients with our model are close to the literature ones (or the ones we measured with the dedicated experiment) confirms both: ability of the model to capture the whole picture of exciton dynamics and validation the usage of literature values. The following statement has now been added in the main text to make our approach clearer:

- *"We want to highlight that the fitted values in Table 1 are close to the experimental and literature values of rate constants that were used as the starting values for the fit, thus confirming accuracy of our model."* [page 7]

The statement in the last second sentence in "Discussion" section: "The electrical model predicts threshold of F8BT and PFO to be around 105 A cm^{-2} and $3 \times 10^3 \text{ A cm}^{-2}$ respectively, which is well below the experimentally achievable current densities ($\sim 200 \text{ A/cm}^2$)."

It is confused this sentence, below or above?

Reply: We thank the Reviewer for pointing out this. The logical error has been fixed in the text. In addition, more precise threshold values have been added with the final texts as following:

“The electrical model predicts thresholds of F8BT and PFO to be around 7.7×10^4 and 2.2×10^3 $A\ cm^{-2}$, respectively (indicated by the red line), which is well above the experimentally achievable current densities (≈ 200 $A\ cm^{-2}$).” [page 13]

As a typical OLED for further pulse investigation, except for the current density and brightness, other direct characterization data are also crucial, such as the stability of the devices under high current, if there is fast decay for the brightness?

Reply: We agree with the Reviewer that the stability of the device is one of the crucial parameters, defining the future potential of our approach. We hence performed an additional stability experiment, where the relative electroluminescence of SY OLEDs was measured as a function of number of applied high current pulses (at three different current densities). The results are shown in the new **Fig. S3**.

Fig. S3 Relative electroluminescence of the SY based OLEDs as a function of the number of applied high voltage pulses (pulse width is 300 ns). Over 90% of the signal remains even after one million pulses at $120\ A\ cm^{-2}$.

Even at the very high current density of $120\ A\ cm^{-2}$ combined with relatively long pulses of 300 ns, any observable drop in performance occurs after over 10^5 pulses and the less than 10% drop in EL after 10^6 pulses indicates excellent stability. The following paragraph has been added in the main text:

- *“Such choice of voltage range and pulse width is dictated by the scope of this work. Staying in this voltage range provides strong current and brightness signal, yet it does not have a significant effect on device degradation (Fig. S3).” [page 4]*

During the pulse measurement, how about the device performances and working conditions? The corresponding I-V curves should provided.

Reply: We have now included the J-V and current efficiency (CE) curves combining DC and pulse data for SY, F8BT and PFO as the new **Fig. S16**. The following statement have also been added in the text:

- “To complete the picture, we included the current – voltage and current efficiency data for F8BT, PFO and SY (as reference) in the form of **Fig. S16**. It combines both the DC and pulse regimes to show full evolution of device characteristics.” [page 12]

Fig. S16. Current-voltage (left) and current efficiency (right) characteristics of OLEDs collected in DC and pulse regimes. (a, b) for SY, (c, d) for F8BT, and (e, f) for PFO.

The authors mentioned “Electrically pumped organic lasers.... the formation of excitons are all strongly non-linear and occur on timescales ranging from sub-ns to μ s range”. References are needed here to prove this point.

Reply: We thank the Reviewer for the valuable comment. In order to illustrate our point better, we have included a new **Fig. S7** for time of maximum exciton generation (and interval where it is over 50% of the maximal value) as a function of injected current density and organic layer mobility.

Fig. S7 Simulated peak exciton formation (charge recombination) time as a function of injected current density and mobility of the emissive layer. **a** At 1/10th of SY mobility. **b** At SY mobility. **c** At 10 times SY mobility. Red shade indicates time interval where the recombination rate is at least 50% of the maximum value for a given current density.

The plot above clearly illustrates that for high mobility of the organic layer and/or large injection current, the time window for carrier recombination can occur as fast as sub-ns (most likely for emitters with high mobility, since current densities of >10⁴ A cm⁻² would be very hard to achieve

experimentally). On the other hand, for a material exhibiting lower mobility and/or lower injection current the peak carrier recombination occurs at tens or hundreds of nanoseconds. Additionally, the peak of recombination is less profound, with significant number of excitons being formed afterwards in the μs time range (red shaded tail). Experimentally speaking, most groups working on organic lasers concentrate on using materials with high mobility with high current injection as possible, thus aiming at the faster end of the figure above. Still, in some older reports, EL response deep in the hundreds of ns is reported (example reference added in the introduction)². Another way to look at the slower end of the spectrum is to think of our experiment as a reverse time-of-flight, with current pulse being the pump and observed EL being the response. In TOF experiments, dealing with less mobile materials and lower current densities, the responses are typically recorded in the μs regime or slower^{3,4}. We have added these references in the introduction. Additionally, we have added the following statements in the main text to point at **Fig. S7**:

- *“We have also simulated the time when the peak exciton formation (charge carrier recombination) takes place as a function of injected current and emitter mobility to illustrate its temporal position and width (Fig. S7). It can be seen that, depending on the material performance and driving conditions, it can be as fast as deep in the sub-nanosecond regime (with a well-defined peak position) or as slow as hundreds of nanoseconds (with the tail extending to μs regime).” [page 7]*

Reviewer #3 (Remarks to the Author):

In this manuscript, Viqar A. et al. tried to study the charge and exciton dynamics of organic light-emitting diodes under high voltage nanosecond pulse: towards injection lasing, a very interesting topic in the field. The authors performed short pulse voltage to understand mechanism of exciton and charge carrier dynamics of OLEDs at high current density. In this work were well done, and the analysis of the results gives physical understanding of devices. Since this study is step for organic semiconductor material and device design for injection lasing, this reviewer would like to recommend for publication in Nature Communications, after considering the following comments.

Reply: Thank you for your positive comments.

Page-2, Abstract do not suffice to understand their role of charge and exciton dynamics of OLED at high current density. “in a time scale spanning from the sub-ns to microsecond region” There is no any explanation or experimental data for microsecond region. “200 A cm⁻² and 2 million cd m⁻²”, Base on author’s conclusion its necessary to applied several K A cm⁻² to get lasing. Why didn’t Author try very short voltage pulse ~ 20 ; In that case they can apply high current to device.

Reply: For the first part of the comment regarding time-scale of exciton generation, please see the answers provided to the same question of Reviewer 2. When it comes to the second part of the comment, the decision to focus on longer pulses at the relatively lower voltage (up to 100 V range) was dictated by the main scope of this manuscript – developing a complete model to describe electrical and exciton dynamics. Within short (i.e. 20 ns) pulse window proper identifying and modelling different exciton pathways would be almost impossible and could easily lead to overfitting of the model. Another reason is the fact that we didn’t have a resonator structure incorporated in our devices, therefore, even pushing the device to the limits it would not probably change emission characteristics. Moreover, in order to build a successful model, we needed to ensure stability of the

devices (see also the comment of Reviewer 2 and the new **Fig. S3**). Finally, with the very short pulse effects like circuit ringing increase in magnitude, altering the results. Nevertheless, as an additional experiment we subjected SY OLEDs to 15 ns pulses of magnitudes up to 450 V. The corresponding current transients are included as **Fig. S4**. The following paragraph has been added in the main text:

- *“300 ns pulse width is a time window long enough to enable simulation of exciton dynamics. Nevertheless, with the future work of reaching injection threshold in mind, we tested capabilities of our setup with shorter pulses (15 ns) of higher magnitude (up to 450 V) – **Fig. S4**” [page 5]*

Fig. S4 Current response of the SY OLED driven with 15 ns pulses of voltages ranging from 30 V to 450 V. **a** Transient current density response, having ringing effects due to short pulses. **b** Plot of the peak current density versus applied voltage for this device.

Page 3, "at high current density of 200 A cm⁻² with a pulse duration of 300 ns" Prof. C. Adachi group reported good approaching for electrical pumping organic lasing base on BSBCz with very high current

density ca. 3 K A cm^{-2} using short voltage pulse (400 ns) (Appl. Phys. Lett.106 093301, 2015 and J. Appl. Phys. 118, 155501, 2015). It is better to compare their work, what is superior of this work comparing with Prof. C. Adachi group reported works? Could Author have try such high current to super yellow, F8BT and PFO based OLEDs?

Reply: This is a good suggestion. Please see the answer to the comment about Page 2, where we included high V transients for SY. Also, please see the answer to the second comment from Reviewer 1, where we show our model accurately reproduces threshold obtained by Prof. C Adachi's group.

Page 4, "The experimental details of device fabrication and the test setup can be found in Supplementary Information" this reviewer couldn't find any device fabrication information in SI

Reply: We thank the reviewer for picking this up, the "Supplementary Information" has been changed to "Materials and Methods section and Supplementary Information" in the text

Page 12, "The electrical model predicts threshold of F8BT and PFO to be around 105 A cm^{-2} and $3 \times 103 \text{ A cm}^{-2}$ respectively, which is well below the experimentally achievable current densities ($\sim 200 \text{ A/cm}^2$)" I am not sure this is correct, Author should check carefully values

Reply: We thank the reviewer for spotting this logical error, it has been fixed in the text (please see reply to reviewer #1)

Page 12, It is well known, for Laser diodes based on organic semiconductor materials have high threshold current densities that require the suppression of various inherent loss processes. One way to study such loss processes is to analyze the EQE roll-off in OLEDs. Author should discuss EQE roll-off behavior of their device.

Reply: We included the J-V and current efficiency (CE) curves for all three materials as a new **Fig. S16**. A clear roll-off in efficiency can be observed there. In addition, in order to identify processes responsible for the roll-off, we used our model to simulate contributions of exciton decay pathways vs injected current density for those three materials – **Fig. S17**. It can be seen that the origin of the efficiency roll-off is strongly material dependent with triplet losses dominating in SY and singlet-singlet annihilation in F8BT. In PFO both SSA and STA loss pathways have impact of similar magnitude. We have added the following passage in the main text:

- *"Additionally, since the efficiency roll-off is evident in **Fig. S16**, we simulated the contributions of different exciton decay pathways as a function of injected current density to identify main loss mechanisms in these materials – **Fig. S17**. One can see that while for SY the dominating loss mechanism is singlet-triplet annihilation, for F8BT it is singlet-singlet annihilation. In PFO both mechanisms show impact within the same order of magnitude."* [page 12]

Fig. S17 Simulated relative contributions of different decay pathways in the total loss of singlet population as a function of injected current density. **a** Main processes responsible for EQE roll-off are STA in the case of SY. **b** SSA for F8BT. **c** A mix of both for PFO. Unsurprisingly, bimolecular processes are dominant at higher current densities.

References

- 1 Sandanayaka, A. S. D. *et al.* Indication of current-injection lasing from an organic semiconductor. *Applied Physics Express* **12**, 061010, doi:10.7567/1882-0786/ab1b90 (2019).
- 2 Chayet, H., Pogreb, R. & Davidov, D. *Transient electroluminescence under short and strong voltage pulses*. Vol. 3148 OP (SPIE, 1997).
- 3 Haber, K. S. & Albrecht, A. C. Time-of-flight technique for mobility measurements in the condensed phase. *The Journal of Physical Chemistry* **88**, 6025-6030, doi:10.1021/j150668a057 (1984).
- 4 Wallace, J. U., Young, R. H., Tang, C. W. & Chen, S. H. Charge-retraction time-of-flight measurement for organic charge transport materials. *Applied Physics Letters* **91**, 152104, doi:10.1063/1.2798592 (2007).

Reviewers' Comments:

Reviewer #1:

Remarks to the Author:

This paper presents a thorough analysis of the various electronic and optical processes occurring in PLEDs, and demonstrates a successful path for predicting the quantitative role of each of these processes. It should definitely be published, but I am not sure that Nat. Comm. is the proper venue, as I still have serious concerns about any strong impact it might have. Specifically, my concerns are as follows:

1. Despite the authors' rebuttal, one of the simple bottom lines resulting from the myriad reports published over 30 years on PLEDs vs small molecule OLEDs is that the former are inherently inferior to the latter. And the danger of excitons getting too close to the metal cathode in the former is just one reason. Admittedly, in the revised manuscript the authors included results on small molecule BSBCz OLEDs, but these were limited to simulations of BSBCz OLEDs reported in the literature.
2. In the Abstract the authors write "The challenge in analysing these processes concurrently has been the major hurdle in realizing injection lasing." That's not true. The major hurdle in realizing injection lasing is the quenching of excitons by other excitons and by polarons, which are processes that become massive at high current densities, b/c they are bimolecular. Hence this statement is not true and, worse still, misleading.
3. Also in the Abstract the authors write "The experimental data is simulated accurately and results highlight crucial differences between optical and electrical pumping and provide a robust test platform for any organic material." The crucial differences between optical and electrical pumping are so obvious they hardly need any highlighting.
4. Finally, still in the Abstract, the authors write that "The key findings provide essential protocols for material and device design for injection lasing." After so many efforts for so many years, it is abundantly clear that if we ever witness an organic injection laser, it will not be b/c we have the "essential protocols for materials and device design" but b/c the groundbreaking group that will demonstrate it will have thought outside the box, in a way we do not currently imagine. Indeed, one of the justifications for the efforts invested in, e.g., OLETs was the opportunity to largely separate charge carriers from the recombination zone and thus minimize exciton quenching by polarons. And we know how well that worked.
5. While the modeling itself appears to be reasonably comprehensive, there is no mention of losses to surface plasmons, which are significant in devices where the emission zone is close to the metal cathode. This is particularly relevant to PLEDs, where the layer directly following the emissive layer is the ultrathin buffer (7 nm Ba in this paper) followed by the metal (typically Al) cathode. All the more reason why the focus on PLEDs in comparing the modeling and simulations to experimental results is less than convincing.

Reviewer #2:

Remarks to the Author:

The authors have addressed my concerns completely and made corresponding revisions and added additional supporting data, which can be accepted without further revision.

Reviewer #3:

Remarks to the Author:

The revised manuscript is well written and organized. Thus I believe that the revised article should be accepted for publication as it is.

REVIEWER # 1 COMMENTS

This paper presents a thorough analysis of the various electronic and optical processes occurring in PLEDs, and demonstrates a successful path for predicting the quantitative role of each of these processes. It should definitely be published, but I am not sure that Nat. Comm. is the proper venue, as I still have serious concerns about any strong impact it might have. Specifically, my concerns are as follows:

1. Despite the authors' rebuttal, one of the simple bottom lines resulting from the myriad reports published over 30 years on PLEDs vs small molecule OLEDs is that the former are inherently inferior to the latter. And the danger of excitons getting too close to the metal cathode in the former is just one reason. Admittedly, in the revised manuscript the authors included results on small molecule BSBCz OLEDs, but these were limited to simulations of BSBCz OLEDs reported in the literature.

Response: We thank the Reviewer's comments and agree that in the typical areas of OLED application, where factors like high EQE, current efficiency or power efficiency are required, small molecule-based OLEDs have emerged victorious over polymers-based OLEDs. A large part of this success is attributed to two factors – development of new classes of emitters that can utilize triplets (phosphorescent and TADF materials, which are not laser active) and wider selection of available deposition tools (evaporation), which led to more sophisticated device stacks with carrier injection and blocking layers, leading to better charge balance and control of the recombination zone.

However, for *lasing*, so far, the winner is less clear. Lasing studies mostly focus on fluorescent emitters, and often the research considers the properties of the emitter itself as opposed to an entire OLED-type device stack. To demonstrate the relevance of polymer emitters, we surveyed the literature for reports of low threshold ($E_{th} < 10 \mu\text{J cm}^{-2}$) organic emitters, for both amplified spontaneous emission (ASE) and lasing. These results are presented in the new Figure S24 and Table S6. Crucially, we find that champion polymers are as relevant as champion small molecules. We want to reiterate that while the statement "PLEDs are inferior to small molecule OLEDs" is generally true (and is in a large proportion result of two main factors mentioned above), "polymer emitters are inherently inferior to small molecule emitters" is not, especially in the lasing field. Our model concentrates on providing a full description of the emitter itself based on its physical parameters: charge mobilities, rate constants, permittivity, threshold and PL shape. Given those parameters, it is equally viable to analyse the response of polymer and small molecule emitters. Different parameters will need to be measured for each new material but the model framework is universal. Indeed, as mentioned by the Reviewer, we demonstrate the viability of our model *for both emitter types* using our own and others' experimental results. The effect of light outcoupling and the remaining layers in the device stack is not the focal point of the model, however, it is included as a simulation parameter as explained in more detail in reply to comment #5.

Fig. S24. Thresholds of champion lasing materials reported in the literature (Table S6) plotted as a function of emission wavelength (A) and aggregated by emitter type (B).

Table S6 Literature reports on best performing ($E_{th} < 10 \mu\text{J cm}^{-2}$) organic emitters manifesting Amplified Spontaneous Emission (ASE) or lasing.

Active Material	Excitation (nm), (laser type)	ASE (nm)	Threshold ASE	Threshold with cavity	Type of Cavity	Reference
Small Molecules						
difluorene with siloxane groups (monolithic liquid)	337, (nitrogen)	386	$1.4 \mu\text{J cm}^{-2}$	–	–	18
C545T (1%) in mCBP co-doped with ACRXTN (6%)	337, (nitrogen)	535	$0.8 \pm 0.3 \mu\text{J cm}^{-2}$	–	–	19
DABNA-2	337, (nitrogen)	494	$1.6 \pm 0.3 \mu\text{J cm}^{-2}$	–	–	20
BSBCz : CBP (6wt%)	337, (nitrogen)	461	$0.32 \mu\text{J cm}^{-2}$	$0.090 \mu\text{J cm}^{-2}$	140, 280 nm period (1-D, mixed order),	21
BSBCz-CN : CBP (6wt%)	337, (nitrogen)	512	$0.63 \mu\text{J cm}^{-2}$	–	–	22
CzPV-SBF	337, (nitrogen)	474	$0.4 \mu\text{J cm}^{-2}$	–	–	23
CzPV-SBF in CBP (6wt%)	337, (nitrogen)	474	$0.11 \mu\text{J cm}^{-2}$	–	–	23
Oligomers						
heptafluorene+CBP	337, (nitrogen)	446	$0.32 \mu\text{J cm}^{-2}$	–	–	24
Octafluorene	337, (nitrogen)	450 nm	$0.090 \mu\text{J cm}^{-2}$	0.084 nJ cm^{-2}	260 and 130 nm (1-D, mixed order)	25
terfluorene	325	423	$1.3 \mu\text{J cm}^{-2}$	–	–	26
pentafluorene	325	442	$1.2 \mu\text{J cm}^{-2}$	–	–	26
Dendrimers						

truxene star-shaped oligofluorenes	380, (Nd ³⁺ :YA G)	437	–	1.29 $\mu\text{J cm}^{-2}$ (Tr 6-3)	260 nm period (1-D, second order), FF=75%	27
truxene star-shaped oligofluorenes	380, (Nd ³⁺ :YA G)	422 - 473	8.4 $\mu\text{J cm}^{-2}$ (Tr3-3)	2.06 $\mu\text{J cm}^{-2}$ (Tr3-3)	270 nm period (1-D, second order), FF=75%	28
truxene star-shaped oligofluorenes	355, (Nd:YV04)	428 - 453	-	1.1 $\mu\text{J cm}^{-2}$ (T4)	270-301 nm period	29
Pyrene core with fluorene dendrons as antenna	380, (Nd ³⁺ :YA G)	491	28, 25 nJ pulse ⁻¹	0.15, 0.26 $\mu\text{J cm}^{-2}$	320 nm period (1-D), FF=75%	30
Polymers						
VE-PFO (B-phase)	355, (Nd ³⁺ :YA G)	467	–	0.4 $\mu\text{J cm}^{-2}$	254 nm period (1-D, second order)	31
PFO:PFO-EH copolymer (4:1)	390	447	–	0.3 $\mu\text{J cm}^{-2}$	290 nm period (1-D), FF=75%	32
terphenylenevinylene polymer (BBEHP-PPV)	337, (nitrogen)	535	0.33 $\mu\text{J cm}^{-2}$	0.04 $\mu\text{J cm}^{-2}$	Not mentioned	33
DCz-LPh5	375 nm, Nd ³⁺ :YAG	447 nm	10 \pm 2 nJ pulse ⁻¹	0.18 $\mu\text{J cm}^{-2}$	290 nm period (1-D, second order), FF=50%	34
F8DP	390 nm, Nd ³⁺ :YAG	452 nm	-	0.036 $\mu\text{J cm}^{-2}$	140 and 280 nm (mixed order)	35
CPDHFPV	337 nm, nitrogen	513 nm	0.16 $\mu\text{J cm}^{-2}$	-	-	36
CPDHFPV in PVK (5%)	337 nm, nitrogen	514 nm	0.02 $\mu\text{J cm}^{-2}$	-	-	36

2. In the Abstract the authors write “The challenge in analysing these processes concurrently has been the major hurdle in realizing injection lasing.” That’s not true. The major hurdle in realizing injection lasing is the quenching of excitons by other excitons and by polarons, which are processes that become massive at high current densities, b/c they are bimolecular. Hence this statement is not true and, worse still, misleading.

Response: We agree with the Reviewer that the wording might have been a bit unfortunate in this sentence and that bimolecular losses are dominating the loss picture at high pumping fluences. The abstract text has been altered to amend that. The entire new abstract, including correction according to comments #3 and #4, is included below:

“Electrical pumping of organic semiconductor devices involves charge injection, transport, device on/off dynamics, exciton formation and annihilation processes. A comprehensive model analysing those entwined processes together is most helpful in determining the dominating loss pathways. In this paper, we report experimental and theoretical results of Super Yellow (PPV co-polymer) OLEDs operating at high current density under high voltage nanosecond pulses. We demonstrate complete exciton and charge carrier dynamics of SY OLEDs starting from charge injection to light emission, in a time scale spanning from the sub-ns to microsecond region, and compare results with optical pumping. The experimental data is accurately replicated by simulation, which provides a robust test platform for any organic material. The universality of our model is successfully demonstrated by its application to three other active materials (F8BT, PFO and BSBCz). The key findings provide a tool to narrow the search for material and device designs for injection lasing”.

3. Also in the Abstract the authors write “The experimental data is simulated accurately and results highlight crucial differences between optical and electrical pumping and provide a robust test platform for any organic material.” The crucial differences between optical and electrical pumping are so obvious they hardly need any highlighting.

Response: We thank the Reviewer’s comment and agree that the differences between optical and electrical are obvious. However, there are not many examples in the literature (some of them mentioned in the main text) of direct application of simplified optical approach to the experimental results obtained with electrical pumping. Some earlier reports lead errors in estimation of current density for lasing threshold (as highlighted by Fig. 4, S9 in the manuscript). Nevertheless, we decided to alter the abstract according to Reviewer’s suggestion, keeping the detailed discussion in the main text (See amended abstract in reply to comment #2).

4. Finally, still in the Abstract, the authors write that “The key findings provide essential protocols for material and device design for injection lasing.” After so many efforts for so many years, it is abundantly clear that if we ever witness an organic injection laser, it will not be b/c we have the “essential protocols for materials and device design” but b/c the groundbreaking group that will demonstrate it will have thought outside the box, in a way we do not currently imagine. Indeed, one of the justifications for the efforts invested in, e.g., OLETs was the opportunity to largely separate

charge carriers from the recombination zone and thus minimize exciton quenching by polarons. And we know how well that worked.

Response: We agree with the Reviewer that the invention of some completely ground-breaking device architecture (e.g. light emitting transistors) would most likely be the biggest stepping stone towards realisation of injection lasing. However, even using traditional OLED architecture, recent results published by Adachi's group indicate that the breakthrough might be near. We hope that the Reviewer agrees, that no matter the architecture used, full understanding of the charge and exciton dynamics, as well as loss pathways in the emitter is one of the important factors in the design of a successful organic injection laser. Such comprehensive model describing the full material response has been missing in the literature and our manuscript aims at filling that gap. We changed the wording in the abstract to highlight that (See amended abstract in reply to comment #2).

5. While the modeling itself appears to be reasonably comprehensive, there is no mention of losses to surface plasmons, which are significant in devices where the emission zone is close to the metal cathode. This is particularly relevant to PLEDs, where the layer directly following the emissive layer is the ultrathin buffer (7 nm Ba in this paper) followed by the metal (typically Al) cathode. All the more reason why the focus on PLEDs in comparing the modeling and simulations to experimental results is less than convincing.

Response: We like to think of the modelling as following the "life cycle" of charges in an emitting device, with four major stages: (1) injection of current as a response to a voltage pulse; (2) charge recombination and exciton formation; (3) evolution of exciton populations and their decay pathways including light emission; and (4) light outcoupling. While the first three components focus on the emitting layer itself, the fourth one (which includes losses due to Surface Plasmon Polaritons, Waveguided modes etc.) is predominantly determined by the other layers of the device. As our model mostly focuses on the response of the emitting layer, it goes into detail analysing steps (1-3) spatially and temporally. Step (4) is included as a parameter translating the number of photons generated in the emissive layer into the brightness values generated by the device in the simulated transients. The value of this parameter is obtained using the Setfos package to simulate our device stack based on complex refractive indices and PL spectra of the materials and device architecture. In support of reply to comment #1, keeping the device architecture same for all the emitters, results in the relative magnitude of outcoupling channels being almost the same for different emitters (with minor differences coming mostly from the shape of spectrum of emitted light, blue shift resulting in slightly higher outcoupling) as can be seen in Fig. S25, with PFO and BSBCz showing the same outcoupling despite one being polymer emitter and other small molecule. Significant change in the outcoupling channels occurs when the remaining part of the device stack is changed, as indicated by simulation of the device structure reported by Adachi's group. We added the following short section in the SI, to provide that information to the reader:

Section A4 *Light outcoupling*

In order to be able to fit the experimental brightness transients, our model has a parameter linking the number of singlets undergoing radiative decay and the device brightness. The value of this parameter is obtained using the Setfos package to simulate our device stack based on complex refractive indices and PL spectra of the materials and device architecture. As can be seen in Fig. S25, keeping the same device stack for different emitters results in fairly consistent outcoupling channels. We also included outcoupling results for the device reported by Adachi's group, which we simulated in the manuscript.

Fig. S25. Relative magnitudes of light outcoupling channels (with SY based device taken as reference) shown for same architecture devices employing other emitters (F8BT and PFO polymers, and BSBCz small molecules) and BSBCz device of the structure reported by Adachi's group¹⁴.

References

18. Ribierre, J.-C. *et al.* Low threshold amplified spontaneous emission and ambipolar charge transport in non-volatile liquid fluorene derivatives. *Chem. Commun. (Camb)*. **52**, 3103–3106 (2016).
19. Nakanotani, H., Furukawa, T. & Adachi, C. Light amplification in an organic solid-state film with the aid of triplet-to-singlet upconversion. *Adv. Opt. Mater.* **3**, 1381–1388 (2015).
20. Nakanotani, H., Furukawa, T., Hosokai, T., Hatakeyama, T. & Adachi, C. Light Amplification in Molecules Exhibiting Thermally Activated Delayed Fluorescence. *Adv. Opt. Mater.* **5**, 1700051 (2017).
21. Sandanayaka, A. S. D. *et al.* Toward continuous-wave operation of organic semiconductor lasers. *Sci. Adv.* **3**, e1602570 (2017).
22. Mamada, M., Fukunaga, T., Bencheikh, F., Sandanayaka, A. S. D. & Adachi, C. Low Amplified Spontaneous Emission Threshold from Organic Dyes Based on Bis-stilbene. *Adv. Funct. Mater.* **28**, 1802130 (2018).
23. Nakanotani, H. *et al.* Extremely Low-Threshold Amplified Spontaneous Emission of 9,9'-Spirobifluorene Derivatives and Electroluminescence from Field-Effect Transistor Structure. *Adv. Funct. Mater.* **17**, 2328–2335 (2007).

24. Zhao, L. *et al.* Singlet-Triplet Exciton Annihilation Nearly Suppressed in Organic Semiconductor Laser Materials Using Oxygen as a Triplet Quencher. *IEEE J. Sel. Top. Quantum Electron.* **22**, 26–34 (2016).
25. Kim, D.-H. *et al.* Extremely low amplified spontaneous emission threshold and blue electroluminescence from a spin-coated octafluorene neat film. *Appl. Phys. Lett.* **110**, 23303 (2017).
26. Choi, E. Y. *et al.* Photophysical, amplified spontaneous emission and charge transport properties of oligofluorene derivatives in thin films. *Phys. Chem. Chem. Phys.* **16**, 16941–16956 (2014).
27. Lai, W.-Y. *et al.* Enhanced Solid-State Luminescence and Low-Threshold Lasing from Starburst Macromolecular Materials. *Adv. Mater.* **21**, 355–360 (2009).
28. Wang, Y. *et al.* Broadly tunable deep blue laser based on a star-shaped oligofluorene truxene. *Synth. Met.* **160**, 1397–1400 (2010).
29. Tsiminis, G. *et al.* Low-threshold organic laser based on an oligofluorene truxene with low optical losses. *Appl. Phys. Lett.* **94**, 243304 (2009).
30. Xia, R., Lai, W.-Y., Levermore, P. A., Huang, W. & Bradley, D. D. C. Low-Threshold Distributed-Feedback Lasers Based on Pyrene-Cored Starburst Molecules with 1,3,6,8-Attached Oligo(9,9-Dialkylfluorene) Arms. *Adv. Funct. Mater.* **19**, 2844–2850 (2009).
31. Kuehne, A. J. C. *et al.* Sub-Micrometer Patterning of Amorphous- and β -Phase in a Crosslinkable Poly(9,9-dioctylfluorene): Dual-Wavelength Lasing from a Mixed-Morphology Device. *Adv. Funct. Mater.* **21**, 2564–2570 (2011).
32. Yap, B. K., Xia, R., Campoy-Quiles, M., Stavrinou, P. N. & Bradley, D. D. C. Simultaneous optimization of charge-carrier mobility and optical gain in semiconducting polymer films. *Nat. Mater.* **7**, 376–380 (2008).
33. Rose, A., Zhu, Z., Madigan, C. F., Swager, T. M. & Bulović, V. Sensitivity gains in chemosensing by lasing action in organic polymers. *Nature* **434**, 876–879 (2005).
34. Wei, Q. *et al.* A High Performance Deep Blue Organic Laser Gain Material. *Adv. Opt. Mater.* **5**, 1601003 (2017).
35. Karnutsch, C. *et al.* Improved organic semiconductor lasers based on a mixed-order distributed feedback resonator design. *Appl. Phys. Lett.* **90**, 131104 (2007).
36. Lee, T.-W., Park, O. O., Choi, D. H., Cho, H. N. & Kim, Y. C. Low-threshold blue amplified spontaneous emission in a statistical copolymer and its blend. *Appl. Phys. Lett.* **81**, 424–426 (2002).
14. Sandanayaka, A. S. D. *et al.* Indication of current-injection lasing from an organic semiconductor. *Appl. Phys. Express* **12**, 61010 (2019).

Reviewers' Comments:

Reviewer #1:

Remarks to the Author:

This paper can now be published in Nat Comm